# The Role of Inflammatory Cytokines in the Pathogenesis of Colorectal Carcinoma—Recent Findings and Review

**DOI:** 10.3390/biomedicines10071670

**Published:** 2022-07-11

**Authors:** Jędrzej Borowczak, Krzysztof Szczerbowski, Mateusz Maniewski, Adam Kowalewski, Marlena Janiczek-Polewska, Anna Szylberg, Andrzej Marszałek, Łukasz Szylberg

**Affiliations:** 1Department of Perinatology, Gynaecology and Gynaecologic Oncology, Collegium Medicum, Nicolaus Copernicus University in Bydgoszcz, 85-168 Bydgoszcz, Poland; szczerbowskikm@gmail.com (K.S.); mm.maniewski@gmail.com (M.M.); l.szylberg@cm.umk.pl (Ł.S.); 2Department of Tumor Pathology and Pathomorphology, Oncology Centre-Prof. Franciszek Łukaszczyk Memorial Hospital, 85-796 Bydgoszcz, Poland; kowalewskiresearch@gmail.com; 3Department of Clinical Oncology, Greater Poland Cancer Centre, 61-866 Poznan, Poland; janiczek.marlena@gmail.com; 4Department of Electroradiology, Poznan University of Medical Sciences, 61-701 Poznań, Poland; 5Department of Clinical Genetics, University Hospital No 1. Bydgoszcz, 85-094 Bydgoszcz, Poland; aniaszylberg@gmail.com; 6Department of Oncologic Pathology, Medical Sciences and Greater Poland Cancer Center, Prophylaxis Poznan University, 61-866 Poznan, Poland; amars@ump.edu.pl

**Keywords:** inflammation, cytokines, cancer, colorectal, carcinogenesis

## Abstract

The inflammatory process plays a significant role in the development of colon cancer (CRC). Intestinal cytokine networks are critical mediators of tissue homeostasis and inflammation but also impact carcinogenesis at all stages of the disease. Recent studies suggest that inflammation is of greater importance in the serrated pathway than in the adenoma-carcinoma pathway. Interleukins have gained the most attention due to their potential role in CRC pathogenesis and promising results of clinical trials. Malignant transformation is associated with the pro-tumorigenic and anti-tumorigenic cytokines. The harmony between proinflammatory and anti-inflammatory factors is crucial to maintaining homeostasis. Immune cells in the tumor microenvironment modulate immune sensitivity and facilitate cancer escape from immune surveillance. Therefore, clarifying the role of underlying cytokine pathways and the effects of their modulation may be an important step to improve the effectiveness of cancer immunotherapy.

## 1. Introduction

Colorectal cancer (CRC) is the third most common and fourth most deadly malignancy worldwide, characterized by a sequential accumulation of multiple genetic aberrations [1]. Epithelial inflammation has been recently deemed as one of the hallmarks of CRC pathogenesis factors, just next to genetic abnormalities. Inflammation impacts all stages of carcinogenesis, including initiation, promotion, and progression [2,3]. Among other mediators of inflammation, cytokines seem to have a unique but complicated role in driving or preventing malignant transformation. The harmony between proinflammatory and anti-inflammatory factors is crucial to maintaining homeostasis. Wherever the balance is shifted towards either side, CRC initiation may occur [4].

In the context of coexistent inflammation, colorectal cancers can be classified into two main subtypes, namely sporadic or colitis-associated cancer. Sporadic cancer arises without known germline mutations, family history of cancer, or inflammatory bowel disease [5]. It is the most common type seen in practice and accounts for more than 60% of CRC cases. Usually, the progression from adenoma to carcinoma takes over 1–2 decades and presents mostly after 60 years of age [1]. Colitis-associated cancer (CAC) is a type of colorectal cancer whose pathogenesis is associated with long-term inflammation. Its occurrence is preceded by clinically detectable inflammatory bowel diseases (IBD), while the risk of CAC correlates with the time and severity of the active disease. CAC is characterized by the formation of polyploid, diffuse or multifocal, and invasive lesions compared to sporadic CRC [6,7]. The preexisting inflammatory processes seem to affect carcinogenesis by causing cellular stress, limiting immune surveillance, rendering the DNA damage response, and accelerating the acquisition of genomic alterations. Both sporadic and colitis-associated cancers are the most frequently characterized by increasing microsomal and chromosomal instability; however, they differ in timing and frequency of specific alterations [3,8,9,10]. Some of them seem pivotal in the early steps of cancerogenesis. For example, the mutation of p53, found in up to 85% of CACs, occurs as one of the first events in the CAC cascade and is more frequent there than in sporadic cancers [11] (Figure 1).

Sporadic cancer and CAC show a similar density of somatic mutations, contradicting the assumption that inflammation outright increases the number of mutations in CAC when compared to sporadic cancers [12]. Nevertheless, CACs harbor some unique genetic alterations. While the frequency of PIK3CA, BRAF, and SMAD4 mutations was similar in both types of cancer, changes affecting genes responsible for cell mobility, cytoskeleton remodeling, and those required for p53 transcription were more frequent in CACs. Noteworthy, Robles et al. detected the amplification of a region-encoding suppressor of cytokine signaling 1 (SOCS1) in CAC tumors. Since SOCS1 downregulates cytokine signaling, including the antitumor activity of IFN-γ and IL-27, its amplification may limit tumor immunosurveillance and enhance inflammation-driven CAC [11,12].

The uniqueness of the genomic landscape of CAC sheds some light on the link between Crohn’s disease (CD), ulcerative colitis (CU), and colorectal cancer. The cumulative risk of CAC reaches up to 8% in patients with CD and CU at 20 years of disease, while CAC causes the death of 15% of IBD patients [13]. Although those diseases differ in cytokine profile, both are associated with significant dysregulation of cytokine transmission. In CD, the predominant type 1 helper T cells (Th1) induce mostly IL-12, IFN-γ, and TNF-α signaling, but in CU, the Th2 cytokines, such as IL-5 and IL-13, play a major role [14]. Furthermore, both CD and CU have distinctive cytokine profiles. While both diseases show frequent downregulation of IL-7, as well as the overexpression of IL-6, IL-12, IL-18, IL-21, IL-27, and IL-34, the overexpression of IL-17, IL-23, and IL-32 seems more specific for CD [15,16,17,18]. On the contrary, higher levels of 5, IL-13, IL-15, IL-23R, and IL-33 were prevalent in CU [19,20].

That imbalance can disturb the local cytokine network, lead to prolonged epithelial stress, and cause tissue injury. In settings of reduced apoptotic stimuli and enhanced proliferation, the risk of genetic error increases, leading to the initiation of carcinogenesis [2,6]. The disruption in the cytokine-mediated crosstalk between immune and epithelial cells in the large bowel seems to be a major factor driving chronic inflammation and leading to the initiation and progression of large bowel cancers [21]. Epidemic studies seem to partly support the hypothesis that chronic inflammation accelerates the formation of cancer because colitis-associated cancers develop in younger patients, are often localized in the right colon, and are characterized by shorter overall survival [22]. Therefore, clarifying the role of underlying cytokine pathways and the effects of their modulation may be an important step to improve the effectiveness of cancer immunotherapy.

In this review, we present the state-of-the-art regarding the impact of inflammatory cytokines on carcinogenesis in the large bowel, briefly summarize the results of recently conducted trials, and discuss their future perspectives. Each cytokine is described with particular attention to its origin, production, role in inflammation, receptors, signaling pathways, influence on tumor microenvironment, and colorectal cancer evolution.

## 2. Inflammation in the Pathways of Sporadic and Colitis-Associated Colorectal Carcinogenesis

A growing number of studies report that inflammation can influence the dynamics and aggressiveness of still-forming malignancy [2,10,21]. Canonically, there are three main pathways of genetic instability and two morphological sequences of CRC carcinogenesis, which differ in pathogenesis and dynamics [10,23,24] (Figure 1). Around 60–85% of CRCs develop through the “classic” adenoma-carcinoma pathway, which may be a physical manifestation of either the chromosomal (CIN) or microsatellite instability (MSI). While the first step in both is the loss of APC function, in the CIN pathways it is usually the sole mutation, which leads to the formation of early adenoma [23]. The following activating mutation of KRAS, Wnt, or other oncogenes, or the loss of 18q LOH, leads to the appearance of a late adenoma stage, which then transforms into adenocarcinoma as a result of the loss of p53 [10]. In the case of the MSI pathway, adenoma is the result of a simultaneous loss of APC and DNA mismatch repair genes (MMR) failure. Hence, a subsequent BRAF, Bax, or TGFβR mutation pushes the neoplasm to the carcinoma stage [25].

**Figure 1 biomedicines-10-01670-f001:**
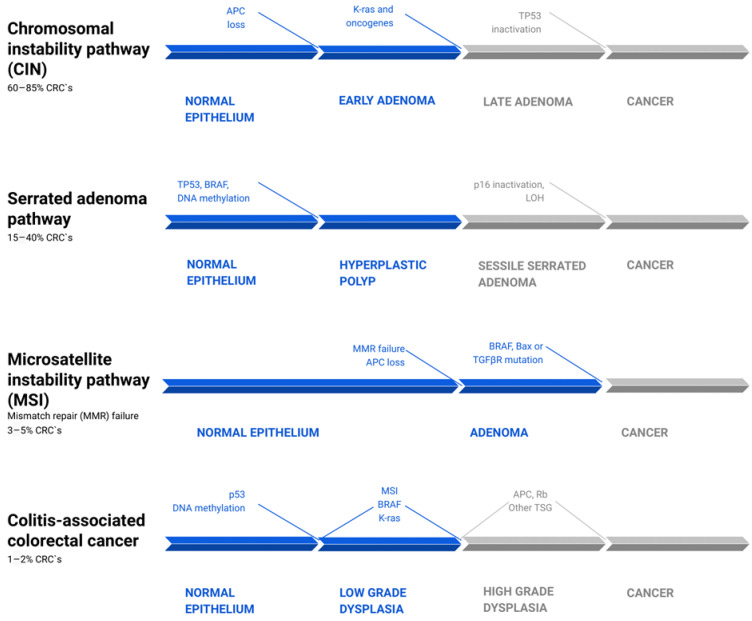
Common pathways of colorectal cancer pathogenesis [10,23,25,26,27]. There are four parallel pathways involved in CRC progression. The chromosomal instability pathway (CIN) accounts for 60–85% of CRC. Mutations, such as APC loss, activation of K-ras or other oncogenes, and then Tp53 inactivation, drive clonal cell growth and ultimately the formation of invasive cancer. The serrated adenoma pathway is responsible for 15–40% of CRC. The occurrence of a BRAF mutation and methylation of cell cycle controlling leads to uncontrolled proliferation of tumor cells. Subsequent methylation of other genes, such as tp53 and p16, promotes the evolution of CRC. Microsatellite instability pathway (MSI) failure leads to 3–5% of CRCs and is driven by simultaneous loss of APC and DNA mismatch repair genes (MMR) failure. The following BRAF, Bax, or TGFβR mutation pushes the tumors from adenoma to CRC. Colitis-associated colorectal cancer accounts for about 1–2% of all CRCs. Its mechanism is similar to that in the pathogenesis of sporadic cancer, including p53, DNA methylation, MSI, BRAF, and K-ras mutations. However, they differ in timing and frequency of specific alterations. These mutations lead to low-grade dysplasia, which, after mutations of APC, Rb, and other TSG, progress to high-grade dysplasia and subsequently to cancer. CRC—colorectal carcinoma; LOH—loss of heterozygosity; TSG—tumor suppressor genes.

The “alternative” serrated pathway, in which 15–40% of CRC develops, is associated with the BRAF mutation and excessive methylation of cell cycle control proteins, which causes the occurrence of serrated adenoma. The pathogenesis of the remaining CRC cases is still not fully explained [28]. In contrast to the pathogenetic pathway of sporadic cancers, colitis-related cancers are better characterized by a “dysplasia-carcinoma” model [8]. In this model, most mutations, including APC and KRAS, occur less often than in sporadic CRC and at later stages of tumor evolution, where they promote cytokine secretion, tumor growth, and angiogenesis via the NF-κB, IL-6/STAT3, or IL-23/Th17 pathways [29,30,31]. APC mutations occur less often and later in CACs than in sporadic cancers, but they often express high levels of β-catenin. Given that APC encodes a negative regulator of β-catenin, suppressing the tumorigenic activity of the Wnt/β-catenin pathway, there is a possibility that chronic inflammation and tissue injury play a major role in the downstream deregulation of Wnt/β-catenin signaling. Those changes may occur even in the absence of APC mutations (Figure 1) [11,12,32].

Most adenomas occur through Wnt-driven transformation of stem cells, while serrated polyps originate from differentiated cells undergoing gastric metaplasia [33]. Importantly, genetically unstable CRCs, outside of the metaplastic lesions, contain adjacent non-metaplastic regions, in which immune cells become exhausted and tumor cells acquire stem-like properties [33]. TGF-β, PI3K, and Wnt signaling alterations are the most frequent in sessile serrated adenoma [34]. 

Reportedly, mutant p53 can drive CAC carcinogenesis by prolonging TNF-α-induced NF-κB activation in cultured cells and increasing IL-6 and IL-8 levels [35]. Inflammation may not be a decisive factor in CRC initiation, and CAC risk correlates with the time and severity of active disease [7]. In the presence of DNA damage response failure, mismatch repair defects, and oncogene-induced replication stress, the coexistent inflammatory process can exacerbate the genetic burden and accelerate tumorigenesis through NF-κB-mediated cytokine production [9]. The semi-local inflammation promotes the accumulation of chemokines, immune cells, stromal cells, and extracellular matrix proteins, which under favorable conditions provide the foundation for the development of the tumor-supportive microenvironment [36]. 

Interleukin-1β, IL-4, and TNF-α are overexpressed in early events of CRC development, such as hyperplastic polyps, adenomas, and serrated adenomas. However, in adenocarcinomas, IL-4 levels are not elevated compared with normal mucosa [37]. The expression of IL-4, TNF-α, COX-2, and IL-1β is higher in serrated adenomas than in adenomas, but the expression of IL-10 shows the opposite trend. Those results suggest that inflammation can be of greater importance in the serrated pathway [38].

## 3. Cytokines, Tumor Microenvironment, and Epithelial–Mesenchymal Transition

In recent years, we have made significant progress to understand the pathophysiology of the tumor microenvironment (TME) and how the initiation of epithelial–mesenchymal transition (EMT) enables cancer cells’ migration and metastasis. Although new reports are still being published, it is already known that interleukins can mediate the crosstalk between cancer cells and the adjacent tissues, impacting carcinogenesis [39]. 

The TME consists of the physical and cellular vicinity of the primary tumor and includes the tumor stroma, tumor-associated cells, immune cells, endothelial cells, macrophages, vessels, and various components of the extracellular matrix (ECM). The immune components of TME are called the tumor immune microenvironment (TIME) and can modulate tumor evolution [39,40]. TIME is composed mostly of myeloid-derived stem cells and regulatory T cells, which mediate local immunosuppression, and cytotoxic T lymphocytes (CTLs), Th2 cells, and macrophages M2, which possess very restricted antitumor activity [41,42]. During the progression of the disease, cancer cells change the properties of surrounding cells, forcing them to produce growth factors and proinflammatory cytokines, such as TNF-α, IL-1β, and TGF-β, that accelerate the acquisition of a mesenchymal-like phenotype during EMT [36].

EMT is a mechanism that allows the cancer cells to transgress their epithelial features and acquire mesenchymal-like properties [43]. The induction of EMT can be triggered by transcription factors, such as ZEB1, whose expression increases with cancer stages and is associated with aggressive disease and poor prognosis in colon cancer [44]. While EMT can occur during wound healing or fibrosis, it is also assumed to be a key player in cancer. As a result of cancer–stromal crosstalk, stationary epithelial cells lose the intracellular adhesive properties, gain spindle cell-like mobility, and become able to migrate from primary tumors [36].

Once the cancer cells exit the bloodstream and form micrometastases, a reverse process, called mesenchymal–epithelial transition, enables them to form metastatic lesions [45]. Tumor-associated macrophages (TAM), myeloid-derived stem cells (MDSC), and lymphocytes Th1/Tc infiltrating the TME influence the epithelial cells undergoing TME through the secretion of IL-1β, IL6, IL8, IL10, IL17, TGF-β, or TNFα [46]. Under the influence of IL-1β, IL-8, and TGFβ, which can affect Wnt/β-catenin signaling, cancer cells can gain stem-like qualities and form a subpopulation of cancer stem cells capable of self-renewal and recreation of the entire tumor population [47]. Simultaneously, the infiltration of suppressive immune cells into the TME increases, and the immunophenotype of local cells changes, leading to the failure of immune surveillance and cancer immune escape [36,39,45,47]. Hence, tumor-associated signaling is essential in understanding the pathogenesis of CRC and in applying potential therapeutic approaches.

## 4. The Mediatory Role of Interleukins in Colorectal Carcinogenesis

Interleukins have gained the most attention due to their potential role in CRC pathogenesis and promising results of clinical trials. They form several families and subfamilies, which have distinct immunomodulatory effects and can cause CRC promotion, tumor growth, and metastasis, or prevent them [48,49]. Malignant transformation is associated with the pro-tumorigenic and anti-tumorigenic interleukins. The harmony between proinflammatory and anti-inflammatory factors is crucial to maintaining homeostasis (Figure 2).

### 4.1. IL-1β

IL-1β is a member of the interleukin-1 family. This cytokine is produced by activated macrophages as a proprotein, which is then proteolytically processed by caspase 1 to its active form. IL-1β can also be secreted by immune, stem, and tumor cells. In that case, it partakes in immunomodulation of TME and can facilitate the immune surveillance escape mechanism [54]. IL-1β mediates cell proliferation, differentiation, and apoptosis, but also induces the proinflammatory response by stimulating the expression of TNFα, IL-6, IL-8, IL-17, COX-2, and PGE_2_ [52,54]. However, the role of IL-1β in CRC carcinogenesis is still not fully explained. Interleukin 1β is overexpressed in both the epithelial and metastatic CRC, and its increased level has been associated with increased CRC growth and invasion [55]. It is most likely caused by the stimulation of metalloproteinase release from the tumor microenvironment, which are the mediators of tissue damage and degradation. IL-1β induces ZEB1, a mediator of EMT formation, and ICAM-1 expression, which promotes adhesion and self-renewal of cancer stem cells (CSC) [54,56,57]. Carcinoma cell-derived IL-1β can then induce PGE_2_ secretion by mesenchymal stem cells (MSC) and enhance paracrine and autocrine signaling, creating a CRC niche [58]. EMT facilitates the acquisition of stem-like properties by cancer cells, represses the expression of E-cadherin, and promotes early metastases. Therefore, upregulation of IL-1β can contribute to the change of TME immunophenotype, the appearance of immunosuppressive tumor niche, cancer cell immune evasion, and facilitate tumor progression [54,56,58]. IL-1β induces the phosphorylation of GSK3β and subsequently increases Wnt/β-catenin signaling, a pivotal step in the initiation of intestinal tumorigenesis [59]. Ping et al. reported that IL-1β upregulates miR-181a, which in turn leads to the downregulation of PTEN. The loss of PTEN is a common occurrence during carcinogenesis and contributes to the increasing genomic instability and cancer growth [60]. IL-1β inhibited BTG anti-proliferation factor 1 (BTG1) expression and promoted the NF-κB pathway in intestinal epithelial cells [48,61]. Tumor cells create a positive feedback loop with macrophages, producing IL-1β and promoting Wnt/β-catenin signaling, which confers drug resistance [62]. The levels of IL-1β in the inflammatory infiltrate in patients with serrated adenoma and adenoma are higher than in normal epithelium and hyperplastic polyps; furthermore, its levels are higher in serrated adenomas when compared to adenomas, suggesting a greater role of IL-1β in the alternative pathway [38]. The expression of IL-1β in M1-like macrophages from left-sided CRC was also higher than from M1 macrophages from right-sided CRC, which is in line with the recent reports regarding the suppression of carcinogenesis-regulating enzymes in distant CRCs [63,64]. Surprisingly, in Haabeth et al.’s study, IL-1β derived from macrophages activated by tumor-specific CD4+ cells was associated with successful cancer immunosurveillance, possibly through enhancing CD4+ T cells’ differentiation and the tumoricidal properties of macrophages [65]. In the human CRC cell line HCT116F, rhIL-1β inhibited the expression of EMT markers and autophagy, migration, proliferation, and invasiveness, while promoting apoptosis of CRC cells [66,67]. Furthermore, while the inactivation of epithelial-specific IL-1R1 resulted in a decrease in CRC burden, a similar knockdown of IL-1R1 in myeloid cells enhanced tumor growth. It seems that IL-1 signaling in myeloid cells can be antitumorigenic and control the local microbiota population, preventing tumor-specific dysbiosis, as well as excessive production of proinflammatory cytokines [66]. Current research suggests that different populations of immune cells stimulated by IL-1β may have opposite effects on tumorigenesis.

### 4.2. IL-4

Interleukin-4 (IL-4) is produced by mast cells, basophils, and activated T lymphocytes. It plays a crucial role in type 2 immune response, differentiation of naive lymphocyte T to follicular and helper type 2, producing antibodies by B cells, expansion of eosinophils and basophils, and skewing of macrophages’ phenotype towards the M2 phenotype [68,69]. Its level is significantly elevated in patients with CRC [70,71]. IL-4 inhibits colon cancer cell–cell adhesion by decreasing the expression of E-cadherin and carcinoembryonic antigen (CEA), but the effect on metastasis and invasion is not clear [38,72]. Kantola et al. reported higher serum levels of IL-4 in CRC patients with distant metastases [73]. IL-4 produced by CD-25+ Th2 tumor-infiltrating cells transforms tumor-associated macrophages into producing high epidermal growth factor (EGF) levels, promoting metastasis [74]. Chen et al. confirmed that the E2F1/SP3/STAT6 route is fundamental in the IL-4-dependent epithelial–mesenchymal transition and CRC cells’ aggressiveness [50]. The overexpression of IL-4 in adenomas, serrated adenomas, and hyperplastic polyps may suggest its role in CRC development [38]. Moreover, Il4/IL4R signaling was found to activate the ERK pathway and impact on the number of osteoclasts in the later stage of CRC metastasis, which accelerates bone destruction [75].

### 4.3. IL-6

Interleukin-6 is a multifunctional cytokine produced mainly by monocytes and macrophages. During inflammatory signaling, IL-6 binds to its receptor, IL-6R, which consists of subunits, the ligand-binding IL6Ra, and the transducing IL-6Rb (gp130), which then activates multiple intracellular pathways through the JAK/STAT pathway [76]. The IL-6 expression level is gradually elevated during the progression from colorectal adenoma to carcinoma, but no clear correlation between IL-6 concentration and risk of polyp number or its type was found [77,78]. Levels of circulating IL-6 are elevated in cancer patients compared to adenoma patients or healthy subjects and correlate with larger tumor size, the occurrence of liver metastases, relapse, and reduced survival in CRC and colitis-associated cancers (CAC) [79,80]. Patients with high IL-6 levels run a greater risk of CRC than patients with low IL-6 levels, independently of disease stage, sex, and age, while the risk seemed proportional to IL-6 level [81]. A systematic review conducted by Veiner et al. showed that IL-6 can predict treatment outcomes in colorectal cancer patients [82]. 

Reportedly, the main causes of elevated IL-6 in colon cancer patients are tumor-associated macrophages, mesenchymal stem cells, and colon cancer-associated fibroblasts (CAF) [29,66]. Paracrine IL-6 activity stimulates tumor evolution, facilitates the EMT, and changes the TME immunophenotype [29,83]. CRC cells augment the secretion of IL-6 and cause the upregulation of integrin αvβ6, enhancing TGF-β signaling and activating CAFs [83,84]. While IL-6 also activates the ERK/MAPK and PI3K pathways in CRC cells, their activation may be dispensable to the IL-6/integrin β6 pathway [83]. Nevertheless, they can promote EMT formation through the STAT3/ERK-dependent activity of the Wnt/β-catenin signaling pathway, M2 macrophage polarization, and promotion of angiogenesis [48,85].

Multiple cytokines, beyond their specific functionality, can also modify the signaling of IL-6. For example, IL-34 can enhance migration and proliferation of CRC cells through ERK1/2 and MAP pathways, cancer cells’ proliferation, and migration by CAFs using netrin-1 and b-FGF induction, but also CRC cells’ diffusion and growth by stimulating TAMs to produce IL-6 [86]. TAM-derived IL-6 drives the crosstalk between MSCs associated with colorectal cancer, enhancing the migration properties of macrophages in TIME, and promoting tumor growth and metastasis through the IL-6/JAK2/STAT3 axis, subsequently activating the Pi3K/AKT/mTOR signaling [87]. The excessive activity of the NF-κB/IL-6/STAT3 axis results in c-Myc and cyclin D1 overexpression, metabolic disorders, tumor growth, progression, and chemoresistance acquisition [29,88]. Furthermore, STAT3 binds to both Mcl-1, Bcl-2, VEGF, and Survivin promoters, preventing CRC cell apoptosis and promoting tumor angiogenesis [89]. Conversely, NF-κB and JAK/STAT3 pathways also contribute to the production of IL-6, forming a cancer-associated NF-κB/IL-6/STAT3 positive feedback loop, making cancer a de facto chronic inflammatory disease [41,90]. This loop, called IL-6 Amp, is active in malignant and adjacent cells, induces a chemokine-like effect, and seems to be a link between the inflammatory lesions and TME. It functions downstream of IL-17, TNFα, and IL-1, modulating the activity of EGFR, c-Myc, VEGF, HIF1-α, matrix metalloproteinases, and multiple proinflammatory cytokines [76,91]. Therefore, a local initiation model was proposed, in which local events, such as tissue injury or oncogenic mutations, activate NF-κB and STAT3, cause the amplification of IL-6 signaling in non-immune cells, and promote the secretion of proinflammatory cytokines. Although this concept addresses primarily chronic inflammation, the author pointed to its potential importance in carcinogenesis [91]. 

IL-6 can suppress the immune response and support the tumor immune surveillance escape in the TIME of CRC. This process is driven through the activation of the JAK/STAT pathway in the dendritic cells (DC) that impairs antigen presentation, limiting the antigen-specific response mediated by INF signaling in Th cells [92]. It also limits the activity of CD4+ and CD8+ T cells, repressing the differentiation of regulatory T cells and promoting the differentiation of Th17 cells from naive CD4+ lymphocytes, contributing to the overexpression of pro-cancerogenic cytokines in CRC, such as IL-17A, IL17F, IL-21, and IL-22, as well as their interactions with non-immune cells [40,93,94]. MDSC attenuates CD4+ Th1 cells’ development through IL-6 production, and IL-6 causes DNA mismatch repair defects, promotes angiogenesis and accumulation of myeloid-derived suppressor cells (MDSCs) in tumors, and shifts the Th1/Th2 balance towards Th2 cells [95,96,97,98]. 

### 4.4. IL-7

IL-7 is produced by non-hematopoietic cells and plays a key role in the adaptive immune response. It can reportedly trigger immune reconstitution in cancer patients by decreasing the programmed death ligand-1 (PD-L1) expression on T cells and increasing the activity of CD4+/CD8+ cells. It enhances the anticancer response and recruits effector cells, such as CTLs and natural killer (NK cells) cells, increasing antigen presentation and tumor surveillance [99]. IL-7 seems to have the potential to diminish the immunosuppressive TME by antagonizing Tregs and MDSCs [100]. IL-7 levels were higher in CRC patients that progressed compared to the non-progression group or adjacent tissue and were associated with a negative response to treatment. Elevated IL-7 was also found in metastatic and lymph node-invasive disease, as well as when CRC was localized in the right colon [16,101]. However, in Wei et al.’s study, IL-7 was significantly downregulated in CRC tissues, decreased along with the progression of the disease, and its downregulation was associated with worse prognosis. It suggests that IL-7 mediated the expression of pro-apoptotic proteins Bax and Bcl-xl in immune cells, leading to tumor cell apoptosis and the inhibition of CRC progression [49]. Due to the ambiguity in the mentioned reports and its potential usage in reshaping the TME, IL-7 is still undergoing a thorough investigation.

### 4.5. IL-8

IL-8, also known as CXCL8, is a member of the CXC chemokine family and is located at the 4q12–13 loci. It binds CXCR1 and CXCR2 receptors, activating multiple metabolic pathways and promoting inflammatory responses [102]. IL-8 is produced by multiple cell types, including monocytes, macrophages, neutrophils, leukocytes, and epithelium, as a result of environmental stress such as hypoxia and exposure to other proinflammatory cytokines including TNF-α and IL-1 [103,104]. TNF-α was shown to be one of the most important agents promoting IL-8 production through the NF-kB/PTEFb-related pathway [105].

CXCR1/2 receptor binding results in activation of the PI3K/AKT/mTOR pathway, which leads to the upregulation of the VEGF and increased angiogenesis, providing tumors with nutrients [106]. Furthermore, AKT promotes the activity of NF-kB, which creates a positive feedback loop of IL-8 synthesis [107]. Independently, CXCR1/2 activation promotes MAPK overexpression, resulting in RAF/MEK/ERK pathway activation, which was shown to be crucial for the development of EMT [108]. Furthermore, IL-8 promotes EMT development through the JAK2/STAT3/Snail pathway and downregulation of the E-cadherin synthesis [109]. 

Elevated IL-8 expression changes the tumor microenvironment in favor of metastasis through increasing neutrophil and tumor-associated macrophages’ infiltration [110]. Additionally, recent studies suggest that cancer cells and CAFs can produce IL-8 to further promote growth and metastasis [111]. 

IL-8 is considered an independent adverse prognosis factor in colorectal carcinoma. Moreover, high IL-8 expression correlated with lymphatic and liver metastasis [112]. The prognostic effect of the IL-8 asset was especially accurate in surgically treated individuals and patients undergoing anti-angiogenesis treatment [113]. This, as well as the reports of the elevation in serum IL-8 after surgical treatment, may suggest that IL-8 plays a crucial role in CRC recurrence [114].

### 4.6. IL-10

IL-10 is an anti-inflammatory cytokine with the ability to inhibit the synthesis of other cytokines, such as IL-1, IL-6, IL-8, and TNF-α [115]. It activates the IL10R1 and ILR10R2 receptors that modulate the activation of multiple pathways, including JAK/STAT3, PI3K/Akt/mTORC1, and SOCS3 pathways [116,117,118]. IL-10 seems to have a controversial role in both inflammation and carcinogenesis. Lower IL-10 levels were associated with a higher risk of the disease, but IL-10 levels in adenoma and serrated adenoma were slightly higher than in normal cells [38,119]. The expression of IL-10 differs between CRC grades, correlates with CRC stages, and is higher in well- and moderately differentiated cancers in comparison to poorly differentiated samples [120]. However, some studies suggest that IL-10 serum levels are lower in the control group than in the CRC patients [119]. Surprisingly, patients in the fourth clinical stage of CRC have a higher level of serum IL-10 when compared to lower stages, while a high serum concentration of IL-10 correlates with poor survival of patients with CRC [121,122]. Additionally, IL-10 overexpression was positively correlated with metastasis occurrence [123]. Furthermore, it was found to be a valid predictive factor of CRC recurrence after treatment and was shown to be an independent risk factor of peritoneal recurrence [124,125,126]. However, recent studies revealed that IL-10 can be a protective factor in animal CRC models. Oral administration of IL-10 reduced the number of polyps/+ model ApcMin, whereas T cell-restricted ablation of IL-10 increased the number of polyps by promoting the accumulation of microbes and eosinophils in colorectal tumors [127]. Similarly, IL-10-deficient mice are more susceptible to spontaneous intestinal tumor development compared with wildtype animals [128]. IL-10 seems to partake in the development of CRC, but its role is still ambiguous. 

### 4.7. IL-11

Interleukin-11 is a member of the IL-6 cytokine family and has been shown in bone marrow stromal cells, osteoblasts, fibroblasts, trophoblasts, synoviocytes, chondrocytes, hepatocytes, gastrointestinal epithelial cells, B and T cells, macrophages, and cardiac myocytes, but the primary source is still not clear [129]. Both IL-11 and its receptor are overexpressed in sporadic CRC samples [130]. MDSCs, cancer-associated fibroblasts (CAFs), and cancer cells upregulate IL-11 transcript levels during CRC development by a positive feedback loop between IL-11-secreting fibroblasts and epithelial colon tumor cells [130,131]. IL-11 is driven by the IL-11/IL-11R/GP130 hexameric complex JAK/STAT3 signaling pathway and STAT3-dependent intestinal tumorigenesis even stronger than IL-6. Interleukin-11 can reduce the efficiency and promote resistance to oxaliplatin [132]. STAT3 phosphorylation induces epithelial cells and colon fibroblast activation, which might be associated with epithelial tumorigenesis and invasion of neoplastic cells [131,133]. Moreover, Wang et al. showed that IL-11 plays a key role in the development of a cancer-promoting microenvironment and forces comatose premalignant cells into CRC [134].

### 4.8. IL-17

The IL-17 family is a group of proinflammatory cytokines that consists of six homologous proteins, from IL-17A to IL-17F. They are produced mainly by Th17, a subset of CD4+ T cells, whose differentiation requires TGF-β, IL-6, IL-21, IL-23, IL-1-β, and STAT3 signaling [135]. It seems that despite operating through the same receptor, IL-17A and IL-17F cause contradictory effects on tumor progression. Most studies suggest a pro-tumorigenic role of IL-17A, which is overexpressed in all steps of the adenoma-carcinoma pathway and associated with the severity of dysplasia. Its expression is higher in CRC compared to previous stages of the pathogenetic sequence, and correlates with tumor size, circulating volume of tumor cells, and shortened survival time [135,136,137]. However, in Lin et al.’s study, IL-17 expression was higher in well-differentiated CRC cells, early-stage disease, and patients with longer overall survival [136]. IL-17A participates in tumor angiogenesis by binding to its receptor on vascular endothelium, promoting angiogenesis and inflammatory response, possibly engaging the IL-17-STEAP4-XIAP pathway and contributing to the immortalization of cancer cells [138,139]. It also promotes cancer growth and survival through the secretion of IL-6 by the NF-kB/STAT3 pathway, increasing the expression of IL-6, IL-8, IL-11, and TNF-α [140,141]. Unlike IL-17A, IL-17F may have antitumor properties that can be associated with its reduced ability to activate the signaling of CXCL1, IL-6, CCL2, CCL7, and MMP13 cascades in comparison with IL-17A [142]. In CRCs TME, IL-17 restricts immune surveillance, inhibits the infiltration of CD8+ T cells, recruits MDSCs by inducing G-CSF production, and promotes cancer-elicit inflammation, favoring the formation of cancer-supportive niches [143,144,145]. Nevertheless, the release of IL-17A by CEA-specific T cells after CRC surgery does not seem responsible for the poorer survival of those patients [146]. On the contrary, IL-17F expression seems to be usually downregulated in CRC cells [147,148]. Omrane et al. reported that the wildtype IL17F gene was associated with longer OS in CRC patients [149]. Furthermore, IL-17F-deficient mice developed CRC more often and had elevated VEGF levels compared to wildtype animals. The authors suggested that IL-17F might reduce VEGF levels, inhibiting tumor angiogenesis [147]. In contrast, a recent Chen et al. study linked the overexpression of IL-17F in tumor mucosa to significantly worse relapse-free survival and overall survival of CRC patients. The recombinant human IL-17 group showed signs of increased CRC cell migration, while the treatment with anti-IL-17 antibodies prevented the formation of CRC cell colonies. Interestingly, in the IL-17 group, the expression of E-cadherin was decreased, but vimentin, Snail1, and Twist levels were elevated compared to the anti-IL-17 group. Those results indicate that IL-17F could also promote CRC progression by inducing the formation of EMT, bringing its supposedly anti-tumorigenic role into question [150]. 

### 4.9. IL-21

IL-21 is a member of the same interleukin family as IL-2, IL-4, and IL-7, which possesses a common receptor γ chain and acts through the JAK/STAT pathway. IL-21 is produced by various immune cells, mainly by activated CD4+ Th1 and Th17 cells, NKT cells, but also by CD8+ and follicular T cells [151]. IL-21 regulates the proliferation of CD4+, prevents IL-2-dependent apoptosis in CD8+ T cells, regulates immunoglobulin production, drives the differentiation of B cells, and limits the differentiation of Tregs, enhancing the immune response [152]. Together with TGF-β, IL-21 initiates the differentiation of naive CD4+ cells into Th17 cells, which can aid tumorigenesis [51]. However, some in vivo studies suggest that IL-21 signaling is protective of colon inflammation in mice due to the IL-21R-mediated downregulation of Th1 and upregulation of Th2, Th17, and Treg responses [153]. While IL-21 is assumed to have antitumorigenic properties, pro-tumorigenic activity has also been reported, especially in colitis-associated carcinoma [154]. IL-21 expression in mice induced NK- and T cell-dependent secretion of IFN-γ and tumor-restrictive activity [155]. Local IL-21 secretion prevents the accumulation of Tregs in the TME and enhances the therapeutic effect of transferred T cells, suggesting that local IL-21 may be an alternative to systemic IL-2 in adoptive cell transfer [156]. IL-21-dependent stimulation of functional intra-tumoral CD8+ cells was able to overcome the checkpoint blockade resistance in renal cancer and enhanced the antitumor activity of tumor-infiltrating lymphocytes [157]. Interestingly, while follicular helper T cells promote the antineoplastic activity of CD8+ T cells through IL-21 signaling, and those effects can be negated by the upregulation of PD-1 expression [158]. The combined therapy of half-life-extended IL-21 and PD-1 blockade additively inhibited tumor growth in mouse models, and enhanced the activity of Th1 T cells, as well as the fraction of DC and M1 macrophages in the TME, sustaining the antitumor immune response [159].

On the other hand, the accumulation of proinflammatory Th17 cells is among the suspected causes of bowel inflammatory diseases, and IL-21 is upregulated in ulcerative colitis, Crohn’s disease, and colitis-associated colon cancer (CAC) [152,160]. The dynamics of IL-21 mRNA were recently examined along with the progression of the adenoma-carcinoma pathway. The levels of IL-21 were higher in adenomas and carcinomas than in controls. Furthermore, colorectal cancer patients with higher IL-21 levels had longer overall survival [161]. The knockoff of IL-21 in mice after azoxymethane and dextran sulfate sodium treatment resulted in reduced infiltration of T cells, mucosal damage, and diminished production of IL-6, IL-17A, and TNF-α. Tumors in IL-21-deficient mice were of smaller size and rarer when compared with wildtype (WT) mice. The underlying mechanism may be connected to the reduction of STAT3 signaling activity, but De Simone et al. report that the oncogenic activity of STAT3, NF-kB, and CRC cells’ survival was not directly affected by IL-21 [15,154,162]. The absence of IL-21 in colorectal cancer TME seemed to shift the balance between Th17 and Th1, as well as between macrophages M2 and M1, towards the latter. It also stimulated the secretion of IL-12 and IFN-γ, leading to the activation of tumor cell apoptosis and TME surveillance [163].

### 4.10. IL-22

CD3 + CD4 + IL-22+ colonic innate lymphoid cells are a major source of IL-22 in the intestine [164]. IL-22 signaling is transmitted through a receptor complex composed of the IL-10Rβ and IL-22Rα1 chains, which are upregulated in primary CRC [165]. IL-22 activates STAT1, STAT3, STAT5, Erk1/2, Akt, and p38 MAPK signaling in epithelial cells, increasing their stemness and tumorigenic potential through the DOT1L methyltransferase [166,167]. High serum levels of IL-22 in CRC tissue are predictive of poor patient survival [168]. IL-22 increases the resistance of CRC cells to chemotherapy via STAT3-dependent autocrine secretion of IL-8. The same pathway suppresses apoptosis and promotes proliferation. IL-22’s role in CRC carcinogenesis may be associated with TNF-γ or Helicobacter hepaticus. It indicates, by synergism, the production of nitrogen oxide intermediates (iNOS) that cause DNA destruction [48,169,170]. In transgenic mice, treatment with anti-IL-22 antibodies together with subcutaneous injection of primary CRC cells strongly impaired tumor development and growth [166,167].

### 4.11. IL-23

IL-23 is a proinflammatory cytokine produced mainly by dendritic cells, macrophages, and neutrophils during intestinal inflammation [171]. IL-23 plays an important role in the pathogenesis of inflammatory bowel disease and colitis-associated cancer [172]. Recent studies show that IL23A secretion is independent in intestinal epithelial cells and its overexpression depends on TNF/NF-kB activation [173]. IL-23 signaling reduces Treg cell activation and promotes IL-22 secretion by innate lymphoid cells and IL-17 production by Th17 cells [174]. Its activity drives intestinal inflammation by inducing other proinflammatory cytokines, such as IL-6, IL-17, and IL-22, and therefore promotes tumor cell survival [133]. IL-23 induces rapid de novo carcinogenesis through Thy1 + IL-23R+ innate lymphoid cells’ activation. IL23A and IL12B transcripts are overexpressed in primary CRC tissues and the IL-23 serum level is elevated in CRC patients’ plasma, leading to an increased rate of metastasis via STAT5 pathway activation [30,133]. Recent studies showed that IL-23 overexpression correlates with tumor stage progression [175]. However, IL-23 serum levels did not correlate with tumor recurrence in patients after surgical treatment combined with chemotherapy [176].

### 4.12. IL-33

IL-33 functions as an “alarmin” released upon cellular stress or damage to promote and amplify inflammation [177]. IL-33 binds to the IL-1 receptor accessory protein (IL-1RA) and ST2L, which forms its receptor complex. Activation of IL-33 signaling leads to the activation of TRAF6, MAPK, and IKK/NF-kB pathways [53]. Luo et al. show that IL-33 inhibits colon cancer growth and metastasis; moreover, the authors present that IL-33 acts on CD4+ T cells by a positive feedback loop and influences the antitumor activity of these cells. On the other hand, IL-33 activates tumor stroma to promote adenomatous polyposis and decreases the barrier function of the intestine [178]. The following translocation of bacterial products to normally sterile tissues indirectly induces the production of pro-tumorigenic cytokines, such as IL-6 [179]. Moreover, IL-33 may trigger the production and synergize with pro-angiogenic factors, such as VEGF, which can facilitate CRC progression [180]. Both IL-33 and its receptor, ST2, are expressed mainly on epithelial cells and myofibroblasts in the CRC microenvironment [178]. ST2L expression is significantly lower in CRC tissue than in non-tumor tissue. The signaling pathway connected to IL-33/ST2 may have a role in protecting against carcinogenesis [181]. Therefore, Maywald et al. suggested that CRC development may be reduced by treatment with ST2-blocking antibodies [178]. On the contrary, Eissmann et al. argued that genetic ablation of ST2 enhanced colon tumor development, while administration of IL-33 reduced CRC growth. ST2 deficiency was associated with an increased population of regulatory T cells and suppression of IFNγ, which correlated to a more aggressive disease [182].

### 4.13. IFN-γ

Interferon gamma is a member of a pleiotropic cytokine family, originally known for its proinflammatory and antitumor effects. It is secreted predominantly by activated CD4 and CD8 T cells, lymphocytes γδ, NK cells, and antigen-presenting cells [183,184]. Single-nucleotide polymorphisms were recently associated with different risks of CRC, suggesting the role of IFN-γ signaling in the progression of colorectal tumors [185]. Similarly, the deficiency of IFN-γ or its receptor was reported to promote colorectal carcinogenesis [186]. The IFN-γ receptor consists of two subunits: IFNγR1 is consistently expressed on the surface of almost all cells, while IFNγR2 expression depends on cellular activity and differentiation, determining cells’ sensitivity to IFN-γ stimulation [187]. It seems that cells producing IFN-γ have lower levels of IFNγR2 and are therefore more resistant to IFN-γ antiproliferative properties. On the contrary, cells that are unable to produce IFN-γ express more IFNγR2 and are more susceptible to its effect [188]. Importantly, the local presence of cytokines may induce IFNγR2 knockdown in the presence of IFN-γ, overcoming cellular desensitization. As a result, IFNγR2 can be a key factor during the Th1–Th2 phenotype switch and following immunomodulation of the CRC microenvironment [184,189].

The biological activity of IFN-γ is enabled by Pi3-Akt-dependent phosphorylation of STAT1 through the JAK/STAT pathway. However, STAT-1-independent signaling can occur via STAT3, which competes with STAT1 for binding to the IFN receptor. STAT1 targets SMAD family member 7, and cell cycle regulators, such as c-Myc and cyclin-dependent kinases. Since IL-6 and IL-10 modulate the activity of STAT3, the local activity of the cytokine signaling favors the dominance of a given pathway [183,190]. The JAK/STAT pathway is often associated with pro-tumorigenic activity; however, STAT1, as a tumor suppressor, modifies the immune profile of tumor cells and their response to therapy [191]. Furthermore, IFN-γ selectively induces apoptosis in stem-like CRC cells through the JAK-STAT1-IRF1 pathway. This effect is attributed to a higher expression of IFNγR on the stem cell surface in comparison to other colon cancer cells [192].

IFN-γ signaling activates host immune surveillance, upregulates the major histocompatibility complex (MHC) molecules, increases antigen presentation, and activates NK cells, cytotoxic CD8+ T cells, and CD4+ T cells. Furthermore, interferon gamma enables the macrophages to switch towards a tumoricidal M1 phenotype, increases their responsiveness to toll-like receptor ligands and TNF, as well as repolarizes CRC’s TME [193]. IFN-γ seems to antagonize the activity of IL-10 and TGF-beta signaling pathways [194,195]. Through a positive feedback loop with IL-12, IFN-γ stimulates the proliferation of cytotoxic T cells and Th1 cells, and inhibits the differentiation of Th2 cells and Th17 cells, limiting IL-4 production, possibly via the inhibition of the IL-4/STAT6 pathway [196]. IFN-γ can enhance the expression of p27Kip, p16, or p21, which induces cell cycle arrest, necroptosis, and apoptosis of tumor cells [197]. Kammertoens et al. showed that the impact of IFN-γ on endothelial cells to IFNγ was sufficient to suppress tumor angiogenesis. It occurred via downregulation of VEGF secreted by stromal fibroblasts, causing local ischemia, and slowing tumor growth [198,199].

While the above findings solidify the role of IFN-γ in the host antitumor response, another side of its activity on CRC progression has recently emerged. IFN-γ activity may limit the efficacy of antitumor immune responses, increase tumor cell genomic instability, and favor tumor immunosurveillance escape [184,200]. It can initiate EMT by affecting the macrophages in the tissue microenvironment through the Wnt/β-catenin pathway [201]. Furthermore, prolonged IFN-γ signaling in tumors can upregulate the expression of PD-L1 and CTLA-4, contributing to the PD-L1-dependent and PD-L1-independent tolerance to immune checkpoint blockade, but also radiation and anti-CTLA-4 treatment through a multigenic resistance program [202]. The IFN-γ-dependent PD-1 overexpression seems to occur via the JAK2/STAT1 signaling pathway [203]. Noteworthy, Lv et al. presented contrary results, in which the downregulation of the INF-γ receptor caused resistance to anti-PD1 therapy in a murine CRC model, seemingly associated with the reduced number of tumor-infiltrating lymphocytes [204]. Consequently, Yuan et al. reported that the knockdown of IFN-γ signaling may improve the response to anti-PD-1 therapy. Supposedly, anti-PD-1 antibodies stimulate CD8+ T cells to secrete IFN-γ, which activates DCs, increases IL-12 secretion in the TME, and enhances the antitumor activity of cytotoxic cells [205,206]. 

### 4.14. TGF-β

Transforming growth factor β is a group of inflammatory cytokines, which was shown to play a pivotal role in inflammation and carcinogenesis [207,208]. TGF-β is a group of three ligands, TGF-β1, TGF-β2, and TGF-β3, which activate two receptors, TGFβR1 and TGFβR2. Binding to its receptor results in the activation of multiple pathways, usually divided into two main groups, namely Smad and non-Smad. The non-Smad group consists of PI3K/AKT, Jun N-terminal kinase (JNK), Rho GTPases, MAPK, and RAS/RAF/MEK pathways [209,210]. Recent studies suggest that dysregulation of the TGF-β signaling can have different results depending on the tissue it affects. For example, TGF-β inhibits the growth of epithelial cells [211]. Additionally, TGF-β depletion promotes the formation of the neoplasm, probably through the limitation of TGF-β-related inhibition of c-Myc and other pro-cancerous factors [212]. However, with the advancement of the disease, TGF-β’s pro-tumorigenic factors begin to prevail. It promotes the formation of the EMT through Smad and non-Smad-related mechanisms, being potentially crucial for angiogenesis and distant metastasis [207,213]. The TGF-β expression shows a stepwise increase from normal epithelium to polyp and tumor cells [214]. TGF-β signaling alterations are the most frequent in sessile serrated adenoma, suggesting its role in the serrated adenoma pathway [34]. TGF-β seems to be a negative prognostic factor in CRC that is associated with advanced stage, recurrence likelihood, decreased survival, and correlated with EMT [215,216]. Furthermore, TGF-β1 protects CRC cells from apoptosis via the XAF1-related pathway [217].

### 4.15. TNF-α

TNF-α is secreted mainly by activated mastocytes, macrophages, and tumor cells. It has two membrane receptors, TNFR1 and TNFR2. TNFR1 has a cytoplasmic domain that can either stimulate cell survival and proliferation through JNK, NF-κB, AP-1, and MAPK pathways, or recruit Fas-associated protein with dead domain (FADD) and pro-caspases that lead to apoptosis. The final effect of TNFR1 activation is determined by the crosstalk with other inflammatory pathways. TNFR2 lacks the death domain and mediates cell activation, migration, and proliferation [218,219]. Recently, TNF-α mRNA overexpression has been associated with high-grade CRC, disease progression, and reduced patient survival [220]. Similarly, higher TNF-α serum levels were found in patients with the higher-stage disease, while the early disease was associated with lower TNF-α serum levels [221]. Since the expression of TNF-α is significantly higher in serrated adenomas compared to adenomas, its role in the alternative pathway may be more important [38].

While the role of TNF-induced inflammation seems to be a significant factor in the tumorigenesis of CRC, it also plays a key role in promoting innate and adaptive immune responses [218]. TNF-α stimulates the early phase of the inflammatory response and regulates the production of other cytokines, increasing vascular permeability, and enhancing oncogene activation, angiogenesis, tumor cell invasion, and migration [222]. Interestingly, its pro-tumorigenic properties may rather be invoked by low chronic TNF-α production than by an intensive outburst that activates reactive oxygen species and kills malignant cells [222]. This theory seems in line with the reports of reduced CRC risk after anti-TNF exposure in patients with long-standing colitis [223,224]. Zhao and Zhang pointed out that this effect may occur through tumor-associated calcium signal transduction protein (TROP)-2 upregulation via the ERK1/2 signaling pathway [225].

In the TME, TNF-α is secreted mostly by TAMs and hinders antigens’ presentation and immunological surveillance. It helps create a tumor-supportive TME by upregulating the expression of PD-L1 on target and adjacent cells [218]. Increased TNF-α signaling enhances the pro-oncogenic signaling pathways in epithelial cells through the Wnt/β-catenin pathway and NF-kB pathways, exacerbating the effects of proinflammatory cytokines in cancer-associated inflammation, especially IL-1β and IL-6, and accelerating tumor growth [226]. Cells exposed to TNF showed increased chromosomal instability, a higher count of mutations, and gene amplification. The expression of chemokine receptor 7 (CCR7) in tumor tissue was also enhanced and is associated with the phosphorylation of kinases ERK and p38, which induce tumor migration. The inhibition of the TNF-α/NF-kB pathway by Kanglaite inhibits the EMT and slows tumor progression [227]. The inhibition of TNF blocks the accumulation of beta-catenin mutations, indicating the mutagenic role of TNF. Moreover, TNF is critical for the formation of intestine polyps in the Apc^Δ468^ mice, a genetic model for intestinal cancer.

Interestingly, according to Ba et al., the clinical effect of targeting TNFα may be associated with changing the ratio of soluble TNFα (sTNFα) and transmembrane TNFα (tmTNFα). The levels of sTNF-α are markedly increased in patients with active ulcerative colitis, suggesting its proinflammatory role in chronic inflammation. However, the tmTNF-α/sTNF-α ratio correlated negatively with IL-1β, IL-6, NO, and M1 macrophages, but positively correlated with the infiltration of MDSC, regulatory T cells, and the levels of IL-10, suggesting anti-inflammatory properties of tmTNF-α [228]. In Schiering et al.’s study, anti-TNFα treatment significantly suppressed polyposis in the Apc^Δ468^ mice. TNF-specific antagonist reduces the number and size of tumors in the AOM/DSS model and limits the risk of colitis-associated CRC [174].

### 4.16. GM-CSF

Granulocyte-macrophage colony-stimulating factor (GM-CSF) is a hematopoietic growth factor promoting the differentiation of various immunological cells, including granulocytes, macrophages, and dendritic cells [229,230]. It is mainly produced by the immune system to induce an immunological response, yet recent studies show that it can also be secreted by other cells, such as endothelial cells and fibroblasts [231].

GM-CSF secretion is stimulated by multiple proinflammatory cytokines, such as IL-1, IL-6, and TNF-alpha, whereas anti-inflammatory cytokines, such as IL-4, IL-10, and IFN-γ, can inhibit its activity [230,232,233,234,235,236,237]. Primarily, GM-CSF was considered an anti-tumorigenic factor due to its ability to activate the immune system. However, recent studies suggest that it can be an important factor in promoting EMT and tumor progression [230,238]. GM-CSF upregulates the VEGF signaling, resulting in activation of the JAK/STAT pathway which induces EMT [230,239]. Furthermore, it also activates other tumor-promoting pathways, including MAPK, PI3K, and NF-kB pathways [240]. Nevertheless, the GM-CSF’s influence on tumor pathogenesis is still unclear. It is a negative prognostic factor in lung cancer, whereas it seems to have inhibitory or ambiguous effects on other neoplasms, such as glioma and melanoma [241].

In CRC, GM-CSF promotes EMT and tumor progression [238]. Furthermore, it can increase angiogenesis via a VEGF-dependent pathway [239]. However, patients with GM-CSF overexpression had increased 5-year survival, which suggests a potential inhibitory effect on tumor progression [241]. Arnold et al.’s study showed that patients with high-eosinophil tumor expression and increased GM-CSF levels also had increased CD8+ lymphocytes’ infiltration and an overall better prognosis than low-eosinophil tumor expression [242]. Therefore, the ambiguous function of the GM-CSF in tumor progression may vary depending on the microenvironment of the tumor.

## 5. Diagnostic and Therapeutic Implications

Considering the profound impact of the cytokine network on the carcinogenesis in the large bowel, therapeutic approaches selectively targeting cytokine pathways have recently emerged. The information about recent and still ongoing trials has been collected in Table 1. Drugs targeting cytokine signaling were tested in monotherapy as suppressants/stimulants of cytokine activity, or as coagents. For example, ANAKINRA, an IL-1 antagonist, was also used for the therapy of CRC to minimize the proinflammatory and pro-tumorigenic properties of IL-1 (NCT02090101). Despite many attempts, due to the overlapping effects of cytokines, the inhibition of one factor is usually not enough to achieve a sufficient outcome. Therefore, cytokine modulation therapies are being tested in combination with various anticancer drugs and protocols. NCAGN01876, an antibody against GITR, a member of the TNF receptor superfamily, was tested in combination with Nivolumab and Ipilimumab, in hope of increasing the effectivity of anti-PD1 and anti-CTLA-4 therapy in colorectal cancer (NCT03126110). Similarly, IFN-γ has been recently administered together with 5-fluorouracil, leucovorin, and with or without bevacizumab to assess the safety, tolerability, and efficacy of such a drug regimen (NCT00786643). 

Nevertheless, serum cytokine levels may be a useful tool in predicting CRC patients’ outcomes. Serum IL-6 and IL8 were shown to be negative prognostic factors in CRC and can be used to identify high-risk patients [243,244]. Furthermore, overexpression of certain cytokines may influence the specific type of treatment. Increased serum levels of IL-6, IL-10, and IL-17A correlate with adverse prognosis in CRC patients undergoing adjuvant chemotherapy and may be useful in monitoring the efficacy of therapy [146,243,245]. The downregulation of other cytokines, such as IL-7, also correlated with a worse prognosis [49]. Currently, more cytokines emerge as potential prognostic markers, but the application of most in everyday practice still seems a matter of the future. 

## 6. Final Remarks

Intestinal cytokines are critical mediators of tissue homeostasis, inflammation, and tumorigenesis (Figure 2). Their expression often correlates with clinicopathological features of CRC, including tumor grade, stage, and patients’ overall survival. The crosstalk of tumor cells, endothelial cells, myofibroblasts, and immune cells in the TME affects tumorigenesis, modulates immune sensitivity, and facilitates cancer escape from immune surveillance in the early stages of carcinogenesis (Table 2). On the other hand, antitumor properties of IL-7 were considered a possible approach to reshape colorectal cancer TME, reduce immuno-resistance, and enhance the treatment response. Currently, multiple cytokines are under thorough investigation to use as therapeutic targets or diagnostic markers. While most results suggest that the modulation of cytokine signaling lacks effectiveness in monotherapy, its potential use in co-therapy still seems very promising (Table 1).

## Figures and Tables

**Figure 2 biomedicines-10-01670-f002:**
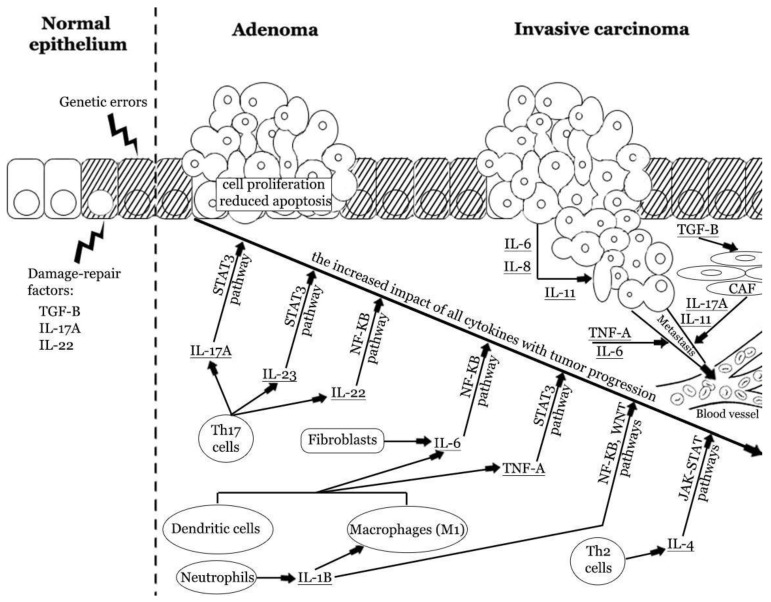
The impact of pro-tumorigenic cytokines in colorectal carcinogenesis. Repetitive colorectal mucosal damage may induce chronic immune system activation and injury of epithelial cells. These processes stimulate cellular augmented proliferation and regeneration, which in turn may result in the accumulation of genetic errors. Inflammatory cytokines may promote tumor formation and enhance progression from adenoma to invasive carcinoma. Cytokines produced by innate and adaptive immune cells and fibroblasts promote cellular proliferation and reduce apoptosis. The latest reports indicate the strong role of IL-17A, IL-6, and TNF-α. These cytokines have an increased impact at different stages of CRC progression. Cancer cells can also produce some cytokines and enhance the vicious cycle of inflammatory response. Finally, cytokines can induce angiogenesis, stromal reorganization, suppression of antitumor immunity, and metastasis [2,29,41,50,51,52,53]. IL—interleukin; Th—T helper cell; TNF-A—tumor necrosis factor α; TGF-β—tissue growth factor β.

**Table 1 biomedicines-10-01670-t001:** Recent studies targeting cytokine signaling in colorectal cancer.

Cytokine	Drug	Type of Intervention	Phase	Study Status	ClinicalTrials.Gov Identifier
TNF	INCAGN01876	Stimulation	I/II	Completed	NCT03126110
TGF	Vactosertib	Inhibition	I	Not yet recruiting	NCT05400122
	NIS793	Inhibition	I	Completed	NCT02947165
	AP 12009	Inhibition	I	Completed	NCT00844064
IL-1	Anakinra	Inhibition	II	Completed	NCT02090101
	CAN04	Inhibition	I/II	Recruiting	NCT05116891
			I	Recruiting	NCT03267316
IL-7	NT-I7	Stimulation	I	Recruiting	NCT04054752
			I	Recruiting	NCT04332653
GM-CSF	GM-CSF	Stimulation	I/II	Recruiting	NCT04929652
	Leukine	Stimulation	I/II	Completed	NCT00785122
	Sargramostim	Stimulation	II	Completed	NCT00103142
		Stimulation	II	Completed	NCT00262808
	JX-594	Stimulation	I	Completed	NCT01469611
	GVAX	Stimulation	I	Recruiting	NCT01952730
IFN-γ	IFN-γ	Stimulation	II	Completed	NCT00786643

**Table 2 biomedicines-10-01670-t002:** Summarized role of cytokines in the pathogenesis of CRC.

Cytokine	Receptor	Impact on Progression	TME Modulation	Main Pathways
IL-1β	TIR	Promotion	Metalloproteinase release [54] EMT promotion [56] CSC promotion [56]	NF-κB/miR-181a/PTEN [60] GSK-3β/Wnt/β-catenin [59]
IL-4	IL-4Rα	Promotion	E-cadherin depletion [38]	ERK [75] E2F1/SP3/STAT6 [50]
IL-6	IL-6R	Promotion	EMT promotion [83] CAFs stimulation [29] Angiogenesis [149] Macrophage migration [149]	JAK2/STAT3 [76] Pi3K/AKT/mTOR [87] NF-κB/STAT3 [90] ERK/MAPK [83]
IL-7	IL-7Rα	Inhibition	CD4+/CD8+ T cells stimulation [99] PD-L1 depletion [99] NK stimulation [99] Tregs inhibition [100] MDSCs inhibition [100]	Apoptotic pathways through Bax and Bcl-xl proteins [49]
IL-8	CXCR1 CXCR2	Promotion	EMT promotion [108] Angiogenesis promotion [106] E-cadherin depletion [109] Neutrophil stimulation [110] TAMs stimulation [110]	PI3K/AKT/mTOR [106] RAF/MEK/ERK [108] JAK2/STAT3/Snail [109]
IL-10	IL-10R1 IL-10R2	Ambiguous	CD8+ T cells stimulation [115] APCs inhibition [102,115] Th17 lymphocytes inhibition [115]	JAK/STAT3 [117] PI3K/Akt/mTORC1 [118] SOCS3 [117]
IL-11	IL-11RA	Promotion	Fibroblast stimulation [131] Epithelial cells stimulation [133]	JAK/STAT3 [132]
IL-17	IL-17RA	Promotion	Angiogenesis promotion [150] MDSCs promotion [144] CD8+ T cells inhibition [144] E-cadherin depletion [150]	NF-kB/STAT3 [246] ERK/MMP 2 and 7 [246] STAT3/VEGF [246] STEAP4-XIAP [138]
IL-21	IL-21R	Ambiguous	CD8+ cells promotion [152] Tregs inhibition [152] Th17 promotion [153] Th1 inhibition [153] Th2 promotion [153]	JAK/STAT3 [151]
IL-22	IL-22R	Promotion	EMT promotion [247]	STAT1,3,5 [167] ERK, Akt, p38, MAPK pathways [167] DOT1L [166]
IL-23	IL-23R	Promotion	Tregs inhibition [174] Th17 promotion [174]	STAT5 [30] TNF/NF-kB [173]
IL-33	IL1RAP sST decoy	Ambiguous	CD4+ T cells promotion [178] Angiogenesis promotion [180]	IL-33/ST2 [53] TRAF6/NF-kB [53] MAPK/AP-1 [53]
IFN-γ	IFNγR1 IFNγR2	Inhibition/ambiguous	Activation host immune surveillance [193] Upregulation the MHC molecules [193] Switch towards M1 and Th1 phenotypes [189,193] EMT promotion [201]	JAK/STAT/IRF1 [192] IL-4/STAT6 [196] Wnt/β-catenin [201]
TNF-α	TNFR1 TNFR2	Ambiguous	Angiogenesis promotion [222] PD-L1 upregulation [218] Epithelial cells promotion [226] EMT inhibition [227]	TROP-2/ERK/p38 [225] NF-κB/STAT3 [90,227] Wnt/β-catenin [226]
TGF-β	TGFBR1 TGFBR2	Ambiguous	Epithelial cells inhibition [207] EMT promotion [213] Angiogenesis promotion [207]	Smad [210] PI3K/AKT [210] RAS/RAF/MEK [210] MAPK [210] JNK [210]
GM-CSF	GM-CSFR	Ambiguous	Activation of immune response [231] EMT promotion [23,230] Tumor progression [230,238] Upregulation of VEGF signaling [230,238] Increase in CD8+ lymphocytes infiltration [242]	MAPK [240] PI3K [240] NF-kB [240]

IL—interleukin; TNF-α—tumor necrosis factor α; TGF-β—tissue growth factor β; IFNγ—interferon gamma; GM-CSF—granulocyte-macrophage colony-stimulating factor; TAM—tumor-associated macrophages; APC—antigen-presenting cells; EMT—epithelial–mesenchymal transition; MHC—major histocompatibility complex.

## Data Availability

Not applicable.

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
