# Peer review of "The Role of Inflammatory Cytokines in the Pathogenesis of Colorectal Carcinoma—Recent Findings and Review"

_biomedicines, 2022, doi:10.3390/biomedicines10071670_

Round 1

Reviewer 1 Report

In this manuscript Authors reviewed the current knowledge about the role of inflammatory  cytokines in the onset and progression of colorectal carcinoma. The review is overall clear and well written, but I noticed that some recent references are missing (e.g. Zhang Q, et al.BMC Gastroenterol. 2021 Jul 12;21(1):284.; Zhou X, et al. Front Immunol. 2021 Jul 23;12:665002; Franzè E et al. Cell Death Discov. 2021 Sep 17;7(1):245.; Zhou X, et al.Front Immunol. 2021 Jul 23;12:665002.) so I suggest Authors to careful check all recent literature or explain the criteria of the articles selection.

Furthermore, I have some suggestions:

-        To define in the introduction the followed scheme in the description of each cytokine.

-        To better describe the characteristics of colitis-associated carcinoma and the differences between CAC and sporadic CRC.

-        A paragraph on IFN-g could be added as an important proinflammatory cytokine, with a role in CRC.

-        As cytokines are important therapeutic target in CRC, I suggest to summarize in a new paragraph all therapeutic approaches targeting selected cytokine pathways or networks and, possibly, clinical trials on  cytokine-modulatory therapies.

Author Response

Dear Reviewer, 
We truly appreciate all your comments and suggestions, and we also believe that your input really helped to improve our work. We addressed all of your suggestions below. I also uploaded a PDF version as an attachment. 

Suggestion 1:The review is overall clear and well written, but I noticed that some recent references are missing, so I suggest Authors to careful check all recent literature or explain the criteria of the articles selection.

Answer 1: We’ve reexamined the latest literature and decided to take into account some references, which were not previously included in this manuscript. Due to recent progress regarding the functionality of tumor microenvironment, colon-associated cancers and the role of cytokine network in colorectal carcinogenesis, we had to include cytokines in which understanding the most progress was made. Given the enormous volume of available data, this choice, by its nature, may be prone to some kind of scientific bias. To minimize its influence, we tried to include information regarding other cytokines within other sections, even if some of them, such as IL-34, doesn’t have .their own paragraphs.

S2: Define in the introduction the followed scheme in the description of each cytokine.

A2: A brief description of the scheme in which cytokine was described has been added at the end of the introduction. The order is: cytokine origin and production, role in inflammation, receptors, signaling pathways, and influence on tumor microenvironment and colorectal cancer evolution.

S3:Describe the characteristics of colitis-associated carcinoma and the differences between CAC and sporadic CRC

A3:Definitions of sporadic CRC and CAC were added to the introduction. Suitable paragraph regarding the differences in genetic alterations and cytokine profile between CAC and CRC were added to eitroduction and the second paragraph “Inflammation in the pathways of sporadic and colitis-associated colorectal carcinogenesis”

S4:  A paragraph on IFN-g could be added as an important proinflammatory cytokine, with a role in CRC.

A4: A paragraph regarding the role of IFNγ has been added. The summary tables have been updated to better reflect the content of the manuscript.

S5: As cytokines are important therapeutic target in CRC, I suggest to summarize in a new paragraph all therapeutic approaches targeting selected cytokine pathways or networks and, possibly, clinical trials on  cytokine-modulatory therapies.

A5: An additional section and a summing up table containing therapeutic approaches and clinical trials were added to the manuscript (Table 2.; paragraph “Diagnostic and therapeutic implications”. 

Reviewer 2 Report

The review entitled „The role of inflammatory cytokines in the pathogenesis of the colorectal carcinoma – recent findings and review” is dealing with the important role of these cytokines in the complex process of initiation and progression of colorectal cancer (CRC) including the process of progression into invasive and metastatic disease.

The review provides a comprehensive overview on how inflammatory cytokines contribute to CRC development and to transformation of adenoma into the malignant, aggressive phenotype of carcinoma. The review introduces the different interleukines and other cytokines, such as TNF-alpha and TGF-beta and discusses in detail their biological roles and implications in the context of CRC. This is put into the context of the different pathways of CRC development from adenoma to carcinoma. The review contains instructive, well-structured figures and a summarizing table. However, the figure legends are too short (in fact, these are only figure titles) to adequately explain the figure content. Further, all important abbreviations should be provided in the figure legends.

The review is well structured and well written, with only few spelling/grammatical errors, which should be corrected accordingly.

In the list of inflammatory cytokines discussed in this review, the mention of IFN-gamma and also of GM-CSF is entirely missing, although these cytokines play important roles in CRC. These should definitely be added to the review to broaden the picture.

One other minor criticism concerns the impact of inflammatory cytokines in chronic diseases of the gut (inflammatory bowel disease), which frequently transform into malignant lesions. Here, impact of such chronic inflammation should have been discussed in little more detail with focus on ulcerative colitis and Crohn’s disease.

Further, diagnostic and also therapeutic implications of inflammatory cytokines for CRC should be reflected in brief – at least as additional paragraph in the Final Remarks section.

Author Response

Dear Reviewer, 

We truly appreciate all your comments and suggestions, and we also believe that your input really helped to improve our work. We addressed all of your suggestions below. 

S1: The figure legends are too short (in fact, these are only figure titles) to adequately explain the figure content. Further, all important abbreviations should be provided in the figure legends.

A1: We added adequate descriptions to explain the content of each figure. The explanations of all important abbreviations have been provided in the figure legends.

S2: The review is well structured and well written, with only few spelling/grammatical errors, which should be corrected accordingly.

A2: The manuscript has been spell checked and proofread before submitting the corrected version. 

S3: In the list of inflammatory cytokines discussed in this review, the mention of IFN-gamma and also of GM-CSF is entirely missing, although these cytokines play important roles in CRC. These should definitely be added to the review to broaden the picture.

A3: Sections regarding the role of GM-CSF and IFN-gamma in the pathogenesis of CRC have been added. The  summarizing tables have been updated accordingly.

S4: One other minor criticism concerns the impact of inflammatory cytokines in chronic diseases of the gut (inflammatory bowel disease), which frequently transform into malignant lesions. Here, impact of such chronic inflammation should have been discussed in little more detail with focus on ulcerative colitis and Crohn’s disease.

A4: The introduction and the section “Inflammation in the pathways of sporadic and colitis-associated colorectal carcinogenesis” have been widened to describe the differences between sporadic and colon-associated cancers, the distinctive cytokine profiles in Crohn’s disease and ulcerative colitis, and outline the sequence of events that leds from the initiation of chronic inflammation to the formation of invasive cancer. However, while we addressed the unique cytokine profile of Crohn’s disease and ulcerative colitis, aforementioned sections focus on the description of chronic inflammation as a continuous, varied process, but do not delve into the detailed differentiation of both diseases.

S5:Further, diagnostic and also therapeutic implications of inflammatory cytokines for CRC should be reflected in brief – at least as additional paragraph in the Final Remarks section.

A5:Additional paragraph regarding clinical approaches and clinical trials were added to the manuscript (“Diagnostic and therapeutic implications”, before the “Final remarks” section). Summary table was provided as well. 

Round 2

Reviewer 1 Report

Authors have very conscientiously replied to all questions  and incorporated changes into the new text.  All these improvements enhanced the quality of the paper.

I only noticed  repetitions at the end of page 7,  paragraph on IL1B, that authors  should correct and control the entire manuscript. 

Author Response

Dear Reviewer,

Thank you again for your feedback. We removed the repetition on page 7 ("paragraph about IL-1B) and checked the manuscript for any possible spelling mistakes and duplicate sentences. We have also made a few stylistic changes that should not affect the content of the article.